# Generalized Flow Matching for Transition Dynamics Modeling

## Abstract

Simulating transition dynamics between metastable states is a fundamental challenge in dynamical systems and stochastic processes with wide real-world applications in understanding protein folding, chemical reactions and neural activities. However, the computational challenge often lies on sampling exponentially many paths in which only a small fraction ends in the target metastable state due to existence of high energy barriers. To amortize the cost, we propose a data-driven approach to warm-up the simulation by learning nonlinear interpolations from local dynamics. Specifically, we infer the kinetic energy or "potential energy" of the system from local dynamics data. To find plausible paths between two metastable states, we formulate a generalized flow matching framework that learns a vector field to sample probable paths between the two marginal densities under the learned energy function. Furthermore, we iteratively refine the model by assigning importance weights to the sampled paths and buffering more likely paths for training. We validate the effectiveness of the proposed method to sample probable paths on both synthetic and real-world molecular systems.

## 1 Introduction

Transition dynamics simulation aims to sample transition paths between two metastable states, which is a fundamental challenge in dynamical systems, stochastic processes, and molecular simulations, with broad applications Bolhuis et al. (2002); Vanden-Eijnden et al. (2010); Aranganathan et al. (2024); Pattanaik et al. (2020); Duan et al. (2023; 2024). The key computational obstacle lies in the rarity of transition events such that it requires long-run molecular dynamics (MD) simulations to go over high energy barriers. In addition, there exists an infinite amount of paths that do not end in the target state.

To address these limitations, early work leverage Markov chain Monte Carlo (MCMC) approaches to mix the path distributions Dellago et al. (2002). However, MCMC-based simulation in such high-dimensional spaces suffer from slow mixing time. Alternatively, later work formulate it as a path integral control problem such that an external control is learned to guide the stochastic process for certain terminal conditions (i.e. arrive at the target in finite time) Holdijk et al. (2024); Yan et al. (2022); Das et al. (2021); Rose et al. (2021); Seong et al. (2024). Nevertheless, the path integral control method is known as a shooting method with high variance (i.e. the probability to hit the target is very low). Recently, a variational formulation of learning Doob's h-transform is proposed, which leads to a collocation method that optimizes Gaussian probability paths Du et al. (2024).

Nevertheless, all previous work require an extensive amount of expensive energy evaluations to sample from the path distributions. In this paper, we study the possibility to find a low-cost approximation of the transition paths distribution. In many studies of rare events in molecular systems, local short-run molecular simulations are used to provide information about transitions Sun et al. (2022). Our approach builds on the idea that even with very short-run local molecular simulations, we can initialize transition paths more effectively than common methods like linear interpolation or heuristic-based method Smidstrup et al. (2014). Specifically, we aim to learn a state cost (or "potential energy") from these short-run trajectories which indicates the likelihood of the states despite they are by no means well-equilibrated.

Building on the recent progress of Schrödinger bridge and flow matching models Liu et al. (2024); Neklyudov et al. (2024); Kapusniak et al. (2024), we formulate the problem as a generalized flow matching problem which introduces an additional potential energy as constraint than the normal flow matching setup Lipman et al. (2023). We solve the generalized flow matching problem by learning a vector field to moving from the distribution of one metastable state to another that minimizes the overall transportation cost. To further improve the quality of the learned path, we apply importance sampling to resample transition paths reweighted by the path probability induced by the true energy function.

We validate the effectiveness of the proposed method on both synthetic data and real-world molecular systems. The results demonstrate that our method can sample high-quality transition paths (close to saddle points) with a significantly less energy evaluations to generate local dynamics data.

## 2 Background

### 2.1 Score & Flow Matching

Score-based generative models or diffusion models Song et al. (2021); Ho et al. (2020) build a generative process from tractable prior distribution $p_T(x)$ (e.g. Gaussian) to complex data distribution $p_0(x) \approx p_{\text{data}}(x)$ where $x \in \mathbb{R}^d$. It can be learned by reversing a forward stochastic process from $p_0(x)$ to $p_T(x)$, as follows:

$$\mathrm{d}x_t = \mathrm{f}(x_t, t)\mathrm{d}t + g(t)\mathrm{d}W_t \tag{1}$$

$$\mathrm{d}x_t = \left[ \mathrm{f}(x_t, t) - g^2(t)\nabla_x \log p_t(x_t) \right] \mathrm{d}t + g(t)\mathrm{d}\bar{W}_t \tag{2}$$

where $\mathrm{f} : \mathbb{R}^d \times \mathbb{R} \to \mathbb{R}^d$ is the drift, $g : \mathbb{R} \to \mathbb{R}$ is the scalar diffusion coefficient, $W_t$, $\bar{W}_t$ are $d$-dimensional Wiener process and the score network $s_\theta(x_t, t)$ is learned to match the conditional score function $\nabla_x \log p_t(x_t|x_0)$:

$$\min_s \; \mathbb{E}_{\mathcal{U}(t)}\mathbb{E}_{p_0}\mathbb{E}_{p_{t|0}} \left[ \|s_\theta(x_t, t) - \nabla_x \log p_t(x_t|x_0)\|^2 \right] \tag{3}$$

Flow matching Lipman et al. (2023); Albergo et al. (2023); Liu et al. (2023) generalizes the idea of score matching by extending it to a general family of Gaussian conditional probability paths $p_t(x_t|x_0) = \mathcal{N}(x_t|\mu_t(x_0), \sigma_t(x_0)^2 I)$ and regress the vector field $v_\theta(x_t, t)$ to the conditional vector field $u_t(x_t|x_0) = \frac{\sigma_t'(x_0)}{\sigma_t(x_0)}(x_t - \mu_t(x_0)) + \mu_t'(x_0)$ corresponding to the Gaussian path $p_t(x_t|x_0)$:

$$\min_v \; \mathbb{E}_{\mathcal{U}(t)}\mathbb{E}_{p_0}\mathbb{E}_{p_{t|0}} \left[ \|v_\theta(x_t, t) - u_t(x_t|x_0)\|^2 \right] \tag{4}$$

### 2.2 Generalized Schrödinger Bridge Matching

One key ingredient behind the success of score-based generative and flow matching models is the simulation-free forward process (diffusion paths or Gaussian paths) that can be evaluated analytically. However, there is a broader class of problems with nonlinear drift or constraint that require simulation of the forward process. The generalized Schrödinger bridge problem is more general such that in addition to learn a stochastic process with minimal kinetic energy that connects $p_0(x)$ and $p_T(x)$, it further minimizes the potential energy $V(\cdot)$ along the path as Liu et al. (2024):

$$\min_{\mathrm{f}, p_t} \int_0^T \int \left( \frac{1}{2}\|\mathrm{f}_\theta(x_t)\|^2 + V(x_t) \right) p_t(x_t)\mathrm{d}x\mathrm{d}t, \tag{5}$$

$$\text{s.t.} \quad \partial_t p_t(x_t) = -\nabla \cdot \left( p_t(x_t)\mathrm{f}_\theta(x_t, t) \right) + \frac{1}{2}g(t)\Delta p_t(x_t), \; p_0 = \mu_0, \; p_T = \mu_T \tag{6}$$

To solve the problem in eq. (5), Liu et al. (2024) propose to approximate the true marginal distribution $p_t$ of the stochastic process with Gaussian density conditioned on both end points $p_{t|0,T}(x_t) = \mathcal{N}(x_t|\mu_t, \sigma_t^2 I)$ where $\mu_t$ and $\sigma_t$ can be parameterized by neural networks or spline. Similar to flow matching, approximating the marginals using Gaussian paths allows for simulation-free sampling and thus easy optimization within

the the objective eq. (5). In the end, $f_\theta$ is learned by matching against the vector field corresponding to the Gaussian path similar to eq. (4). It is worth noting that a similar general framework has also been proposed in Neklyudov et al. (2023; 2024).

## 3 Generalized Flow Matching for Transition Dynamics Modeling

### 3.1 Short-Run Molecular Dynamics

Molecular dynamics simulation follows Newton's equation of motion such that $M\ddot{x}_t = -\nabla_x U(x_t)$, where $M$ is the mass of the particles, $U : \mathbb{R}^{N \times 3} \to \mathbb{R}$ is the potential energy function and $x = (x_1, \ldots, x_N) \in \mathbb{R}^{N \times 3}$ is one state of the molecular system. Nevertheless, one common goal of interest to run molecular simulation is to sample from the thermal equilibrium (known as the NVT ensemble) with a heat bath at a fixed temperature, which is taken into account by running the following Langevin equation.

$$\begin{pmatrix} dx_t \\ dv_t \end{pmatrix} = \begin{pmatrix} v_t \\ -M^{-1}\nabla_x U(x_t) - \gamma v_t \end{pmatrix} \mathrm{d}t + \begin{pmatrix} 0 & 0 \\ 0 & M^{-1/2}\sqrt{2\gamma k_B T_\mathrm{p}} \end{pmatrix} \mathrm{d}W_t \tag{7}$$

where $\gamma$ is the friction coefficient, $k_B$ is the Boltzmann constant and temperature $T_p$. The stationary distribution of running this Markov process is the thermal equilibrium $p(x) \propto e^{-U(x)/k_B T_p}$. However, with the goal of learning transition paths with a limited number of potential energy evaluations, we do not attempt to sample from the equilibrium.

Instead, we consider two predefined metastable states $A$ and $B$, which correspond to local minima on the potential energy landscape. We run only local molecular dynamics simulations around two metastable state to initialize our subsequent methods. In particular, we let $\Phi_{s:t}^{\rightarrow}(y)$ denote simulating the dynamics in eq. (7) initialized at point $y$ and time $s$ for time $t - s$ and $t > s$. For $t < s$, we integrate backward in time and write $\Phi_{s:t}^{\leftarrow}(y)$. For a short time interval $s$ chosen as a design decision, our subsequent methods are initialized from the induced distributions $\mu_0(x) = (\Phi_{-s:0}^{\rightarrow})_\# \delta_A$ and $\mu_T(x) = (\Phi_{T+s:T}^{\leftarrow})_\# \delta_B$ where we abbreviate $\delta_A(y) = \delta(A - y)$, $\delta_B(y) = \delta(B - y)$.

We illustrate examples of $\mu_0$ and $\mu_T$ resulting from different choices of $s$ in fig. 1, where $A$ and $B$ are indicated using green points and $\mu_0$ and $\mu_T$ are indicated using red and blue points, respectively. Our hope is that short-run MD simulations provide a useful initialization for our Generalized Flow Matching transport problem with learned potentials.

### 3.2 Generalized Flow Matching

Inspired by the generalized Schrödinger bridge problem, we formulate the problem of finding feasible transition paths as a distribution matching problem such that we learn a vector field to transport between two given marginal distributions:

$$\begin{aligned} \mathcal{L}_{\mathrm{GFM}} = \min_{v_t^\theta, p_t} \int_0^T \int \left( \frac{1}{2}\|v_t^\theta(x_t)\|^2 + V_t(x_t) \right) p_t(x_t)\mathrm{d}x\mathrm{d}t \\ \text{s.t.} \quad \partial_t p_t(x_t) = -\nabla \cdot \left( p_t(x_t)v_t^\theta(x_t) \right), \ p_0 = \mu_0, \ p_T = \mu_T \end{aligned} \tag{8}$$

where $\mu_0$ and $\mu_T$ are the density of the states explored by a local molecular dynamics simulation around the metastable states $A$ and $B$, described above.

In general, the kinetic energy corresponds to the speed of transport while the potential energy defines a state cost corresponding to how probable the states are (e.g. likelihood). Together these cost functions define an optimal interpolation of marginals $p_t^*$ between $\mu_0$ and $\mu_T$, which reduces to the dynamic formulation of optimal transport for $V_t(\cdot) = 0$ Peyré et al. (2019). It is worth noting that the potential energy function $V(\cdot)$ does not have to be the same as $U(\cdot)$ used in molecular dynamics simulation, and we now discuss how to learn a surrogate potential energy functions from $\mu_0, \mu_T$ to guide the sampling of transition paths.

### 3.3 Inferring Kinetic and "Potential Energy" from Data

As one of the main goal of this study is to reduce the number of potential energy evaluation $U(\cdot)$, we show how we can learn a surrogate energy function $V(\cdot)$ from the local dynamics data in two ways Pooladian et al. (2024).

*Latent interpolation*: We propose to learn an autoencoder that maps high-dimensional data into low-dimensional representations such that it preserves structural information Liu et al. (2024). The hypothesis is that the latent space compresses semantic information from data thus better measure distance than the ambient Euclidean space. Specifically, we map data $(x_0, x_T)$ to the latent space as $(z_0, z_T)$. We define the state cost (or "potential energy") as the deviation of the state $x_t$ from certain interpolation (e.g. spherical interpolation) of $x_0$ and $x_T$ in the latent space $I(z_0, z_T, t)$:

$$V(x_t) = \|x_t - \text{Decoder}(I(z_0, z_T, t))\|_2^2 \tag{9}$$

*Metric learning*: Another natural way to learn the potential energy is through metric learning such that the metric informs how dense the data is around a particular location Arvanitidis et al. (2021; 2018). Once the metric $G : \mathbb{R}^{N \times 3} \to \mathbb{R}^{N \times N}$ is learned, the kinetic energy term in eq. (8) will become Kapusniak et al. (2024):

$$\|v_t\|_G = v_t^T G(x_t) v_t = v_t^T v_t + v_t^T (G(x_t) - I) v_t = \|v_t\|_2 + V(x_t, v_t) \tag{10}$$

where the potential energy is implied as:

$$V(x_t, v_t) = v_t^T (G(x_t) - I) v_t \tag{11}$$

It is worth noting this can be equivalently considered as defining the kinetic energy under a learned metric tensor $G$ because potential energy often only depends on $x_t$, where in the other case, the kinetic energy is defined under the trivial diagonal metric in Euclidean space. We leave the details about metric learning in Appendix A.

### 3.4 Conditional Generalized Flow Matching Objective

To facilitate optimization of eq. (8), we derive the following conditional or 'bridge' objective.

Assume the path of marginals $p_t$ decomposes such that $p_t(x_t) = \int \int p_{0,T}(x_0, x_T) p_{t|0,T}(x_t) dx_0 dx_T$ and there exists $v_{t|0,T}$ such that $\partial_t p_{t|0,T} = -\nabla \cdot (p_{t|0,T} v_{t|0,T})$. We define the following *conditional* GFM objective

$$\mathcal{L}_{\text{cGFM}}(x_0, x_T) := \min_{p_{t|0,T}, v_{t|0,T}} \int_0^T \int \left(\frac{1}{2}\|v_{t|0,T}(x_t)\|^2 + V_t(x_t)\right) p_{t|0,T}(x_t|x_0, x_T) dx_t dt \tag{12}$$
$$\text{s.t.} \quad \partial_t \, p_{t|0,T}(x_t) = -\nabla \cdot (p_{t|0,T}(x_t) v_{t|0,T}(x_t)), \quad p_{0|0,T} = \delta_{x_0}, \quad p_{T|0,T} = \delta_{x_T}.$$

where we abbreviate the boundary condition $p_{0|0,T}(x) = \delta(x - x_0)$ and optimize a conditional objective for each sample from the joint distribution $(x_0, x_T) \sim p_{0,T}$.

The objective in section 3.4 is a deterministic analogue the conditional stochastic control objective in Prop. 2 of Liu et al. (2024). We next show that the optimizing the conditional objective yields an *upper bound* on the marginal objective, which was not emphasized by Liu et al. (2024). See appendix C.2 for proof.

**Proposition 1** *Taking the expectation of the conditional objective over $(x_0, x_T) \sim p_{0,T}$ and enforcing that $p_{0,T} \in \Pi(\mu_0, \mu_T)$ satisfies the boundary conditions yields an upper bound on $\mathcal{L}_{GFM}$,*

$$\mathcal{L}_{GFM} \leq \mathbb{E}_{p_{0,T}(x_0, x_T)} \left[\mathcal{L}_{cGFM}(x_0, x_T)\right] \quad s.t. \quad p_0 = \mu_0, \; p_T = \mu_T. \tag{13}$$

For a learned potential energy (section 3.3), our computational approach seeks to find an approximate solution to the marginal GFM problem (eq. (8)) by representing a coupling $p_{0,T}$ and parameterizing $p_{t|0,T}$

and $v_{t|0,T}^{\phi}$. However, to generate unconditional transition paths from an initial $x_0$ or $x_T$ and facilitate a parameterization of $p_{0,T}$, we also consider learning a marginal vector field $v_t^{\theta}$ in eq. (8).

**Neural Spline Parameterization of $v_{t|0,T}^{\phi}$** The conditional objective in section 3.4 requires $p_{t|0,T}$ and $v_{t|0,T}^{\phi}$ which satisfy the continuity equation and respect the boundary conditions $\delta_{x_0}$, $\delta_{x_T}$. To satisfy these desiderata, we use neural networks to parameterize a spline interpolation in the sample space that satisfies the boundary conditions:

$$x_t^{\phi} = (1-t)x_0 + tx_T + t(1-t)\text{NN}_{\phi}(x_0, x_T, t) \tag{14}$$

This spline induces a path of marginal distributions $p_{t|0,T}$, where the velocity $v_{t|0,T}$ satisfying the continuity equation in eq. (12) is directly tractable as Albergo et al. (2023); Tong et al. (2023)

$$v_{t|0,T}^{\phi}(x_t) = x_T - x_0 + t(1-t)\dot{\text{NN}}_{\phi}(x_0, x_T, t) + (1-2t)\text{NN}_{\phi}(x_0, x_T, t) \tag{15}$$

This choice of $v_{t|0,T}^{\phi}$ thus satisfies both the endpoint constraints in eq. (14) and the conditional continuity equation constraint for the marginals $p_{t|0,T}^{\phi}$ induced by $x_t^{\phi}$ sampling. Define $\mathcal{L}_{\text{cGFM}}^{\phi}(x_0, x_T)$ as the value of the objective in eq. (12) for possibly suboptimal $x_t^{\phi} \sim p_{t|0,T}$ and $v_{t|0,T}^{\phi}$ which satisfy the constraints, as is guaranteed by our parameterization above. Using eq. (12) and eq. (13), we have

$$\mathcal{L}_{\text{cGFM}}(x_0, x_T) \leq \mathcal{L}_{\text{cGFM}}^{\phi}(x_0, x_T), \quad \mathcal{L}_{\text{GFM}} \leq \mathbb{E}_{p_{0,T}}\left[\mathcal{L}_{\text{cGFM}}^{\phi}(x_0, x_T)\right] \ \forall p_{0,T} \in \Pi(\mu_0, \mu_T). \tag{16}$$

Using this parameterization, we can tractably optimize the objective $\mathcal{L}_{\text{cGFM}}^{\phi}(x_0, x_T)$ as a function of $\phi$. Expanding to write an expectation over samples from $p_{0,T}$, we minimize the following the objective over $v_{t|0,T}^{\phi}$,

$$\begin{aligned} \mathcal{L}_{\text{spline}}^{p_{0,T}}(\phi) &= \mathbb{E}_{p_{0,T}}\left[\mathcal{L}_{\text{cGFM}}^{\phi}(x_0, x_T)\right] \\ &= \mathbb{E}_{p_{0,T}(x_0,x_T)}\left[\mathbb{E}_{\mathcal{U}(t)}\mathbb{E}_{p_{t|0,T}(x_t)}\left[\frac{1}{2}\|v_{\phi}(x_0, x_T, t)\|_2^2 + V(x_{\phi}(x_0, x_T, t))\right]\right] \end{aligned} \tag{17}$$

where we have replaced the integral over $\int_0^T (\cdot)dt = \mathbb{E}_{\mathcal{U}(t)}[(\cdot)]$ with an expectation over the uniform distribution. In practice, we sample time points $t \sim \mathcal{U}(t)$ during training and use an empirical expectation over $(x_0, x_T) \sim p_{0,T}$.

**Marginal Vector Field $v_t^{\theta}$** To perform unconditional sampling, we need to learn a marginal vector field $v_t^{\theta}$ which simulates the marginals $p_t$ induced by a particular $p_{0,T}$, $p_{t|0,T}^{\phi}$, and $v_{t|0,T}^{\phi}$. This also corresponds to translating a candidate solution for the objective $\mathbb{E}_{p_{0,T}}[\mathcal{L}_{cGFM}^{\phi}(x_0, x_T)]$ in eq. (13) into a solution to the marginal problem in eq. (8). Following similar arguments as Lipman et al. (2023); Albergo et al. (2023); Tong et al. (2023), one can show that the marginal vector field

$$v_t^{(p_{0,T},\phi)}(x_t) = \mathbb{E}_{p_{0,T}(x_0,x_T)}\left[\frac{p_{t|0,T}^{\phi}(x_t)}{p_t(x_t)}v_{t|0,T}^{\phi}(x_t)\right] \tag{18}$$

satisfies the continuity equation for the induced marginals $p_t(x_t) = \mathbb{E}_{p_{0,T}}[p_{t|0,T}^{\phi}(x_t)]$, with $\partial p_t = -\nabla \cdot (p_t v_t^{(p_{0,T},\phi)})$. We use the notation $v_t^{(p_{0,T},\phi)}$ to indicate that the appropriate vector field is induced from our current $p_{0,T}$, $p_{t|0,T}^{\phi}$, and $v_{t|0,T}^{\phi}$. See appendix C lemma 1,

We finally can use the flow matching objective to learn an approximate $v_t^{\theta} \approx v_t^{(p_{0,T},\phi)}$ for our current $p_{0,T}$ and $\phi$ Lipman et al. (2023); Albergo et al. (2023); Tong et al. (2023), where sg$(\cdot)$ indicates stop-gradient,

$$\mathcal{L}_{\text{flow}}(\theta) = \mathbb{E}_{\mathcal{U}(t)}\mathbb{E}_{p_{0,T}}\mathbb{E}_{p_{t|0,T}^{\phi}}\left\|v_t^{\theta}(x_t, t) - \text{sg}\left(v_{t|0,T}^{\phi}(x_0, x_T, t)\right)\right\|^2 \tag{19}$$

Note that we amortize this optimization over training steps by alternating updates for $v_t^{\theta}$, $p_{0,T}$, and $\phi$, so that $v_t^{\theta}$ may not exactly match $v_t^{(p_{0,T},\phi)}$.

---

**Algorithm 1** Generalized Flow Matching Algorithm

---

**Require:** data $p_0, p_T$, learned potential energy $V(\cdot)$, neural spline network $\mathrm{NN}_\phi$, velocity network $v_\theta$ (initialization, e.g. $\mu_0 \otimes \mu_T$), replay buffer $\mathcal{B}$, maximum replay buffer size $|\mathcal{B}|$

1: **while** not converged **do**
2:      Sample $x_0 \sim p_0$, $x_T \sim \mathrm{ODE}([0,T], x_0, v_\theta)$, $t \sim \mathcal{U}(0,T)$
3:      $x_t^\phi = (1-t)x_0 + tx_T + t(1-t)\mathrm{NN}_\phi(x_0, x_T, t)$          eq. (14)
4:      $v_{t|0,T}^\phi = x_T - x_0 + t(1-t)\dot{\mathrm{NN}}_\phi(x_0, x_T, t) + (1-2t)\mathrm{NN}_\phi(x_0, x_T, t)$          eq. (15)
5:      $\mathcal{L}_{\mathrm{spline}}^{p_0,T}(\phi) = \frac{1}{2}\|v_{t|0,T}^\phi\|^2 + V(x_t)$          eq. (17)
6:      $\mathcal{L}_{\mathrm{flow}}(\theta) = \|v_t^\theta(x_t) - \mathrm{sg}(v_{t|0,T}^\phi(x_0, x_T, t))\|^2$          eq. (19)
7:      Update $\phi, \theta$ using $\nabla_\phi \mathcal{L}_{\mathrm{spline}}(\phi)$, $\nabla_\theta \mathcal{L}_{\mathrm{flow}}(\theta)$
8:      **if** Replay Buffer **then**
9:          $\{x_0^i\}_{i=0}^N \sim p_0$
10:        $\{\gamma^i\}_{i=0}^N = \mathrm{ODE}([0,T], \{x_0^i\}_{i=0}^N, v_\theta)$, $\gamma = x_{0:T}$
11:        Compute weight $\tilde{w}^i$ for $\gamma^i$, add $(\gamma^i, \tilde{w}^i)$ to $\mathcal{B}$          eq. (20)
12:        **while** Replay Buffer not converged **do**
13:          Sample $\gamma \sim \mathcal{B}$, $t \sim \mathcal{U}(0,T)$, $x_0 \sim p_0$, $x_T \sim \mathrm{ODE}([0,T], x_0, v_\theta)$
14:          $\mathcal{L}_{\mathrm{replay}}(\phi) = \|x_\phi(x_0, x_T, t) - \gamma_t\|^2 + \|v_\phi(x_0, x_T, t)\|^2$          eq. (21)
15:          $\mathcal{L}_{\mathrm{flow}}(\theta) = \|v_t^\theta(x_t) - \mathrm{sg}\left(v_{t|0,T}^\phi(x_0, x_T, t)\right)\|^2$
16:          Update $\phi, \theta$ using $\nabla_\phi \mathcal{L}_{\mathrm{replay}}(\phi)$, $\nabla_\theta \mathcal{L}_{\mathrm{flow}}(\theta)$
17:        **end while**
18:      **end if**
19: **end while**
20: **return** $\mathrm{NN}_\phi, v_\theta$

---

**Coupling Parameterization**   Finally, the above conditional objectives require samples from a coupling distribution $p_{0,T}$, which should satisfy the boundary constraints $p_0 = \mu_0$, $p_T = \mu_T$ in eq. (13). Note that proposition 1 requires optimization over $p_{0,T}$ and our initial data from MD simulation is unpaired $(p_{0,T} = \mu_0 \otimes \mu_T)$. Thus, optimizing the neural spline on fixed, independent coupling will not lead an optimal solution. We introduce two ideas for maintaining and updating couplings $p_{0,T}$.

*Minibatch Optimal Transport.* Following Pooladian et al. (2024); Tong et al. (2023), we can consider solving for (entropic) optimal transport couplings over an empirical batch of samples $(x_0, x_T) \sim \mu_0 \otimes \mu_T$. However, the choice of cost is crucial to defining the solutions. While the objective in eq. (8) corresponds to dynamical OT with the squared error or Euclidean cost for $V(\cdot) = 0$, calculating the appropriate dynamical cost with nonzero potential energy is an optimization problem in its own right (Pooladian et al., 2024). To simplify the algorithm, we use OT couplings with the Euclidean cost and introduce resampling based on a replay buffer.

*Rectified Flow.* Following Liu et al. (2023); Shi et al. (2024), an alternative technique to solve for optimal transport couplings is to iteratively simulate the marginal vector field $v_t^\theta$. For example, we could sample $p_{0,T}^\theta$ by sampling an initial $x_0 \sim \mu_0$ and simulating to obtain $x_T$. For the Euclidean cost or a further family of convex costs, iterative simulation and matching $v_t^\theta$ approaches the solution to the optimal transport problem Liu (2022). However, again, since our GFM problem involves a dynamical cost, we simplify by using the rectified coupling induced from $v_t^\theta$ and introduce corrections via resampling.

*$\alpha$-DSBM.* Following De Bortoli et al. (2024), we reduce the additional training cost introduced by reflow by employing an online procedure such that we conduct the reflow operation in every training iteration instead of training until converge before each reflow operation. Even though the online procedure improves the training efficiency, it can lead to challenges such as error accumulation thus slow convergence. We adopt the bidirectional reflow procedure such that we train our models with two couplings $p_{0,T}^{\theta,\rightarrow}$ and $p_{0,T}^{\theta,\leftarrow}$ where we sample an initial $x_0^\rightarrow \sim \mu_0$ and simulate to obtain $x_T^\rightarrow$ and conversely we sample an initial $x_T^\leftarrow \sim \mu_T$ and simulate reversely to obtain $x_0^\leftarrow$.

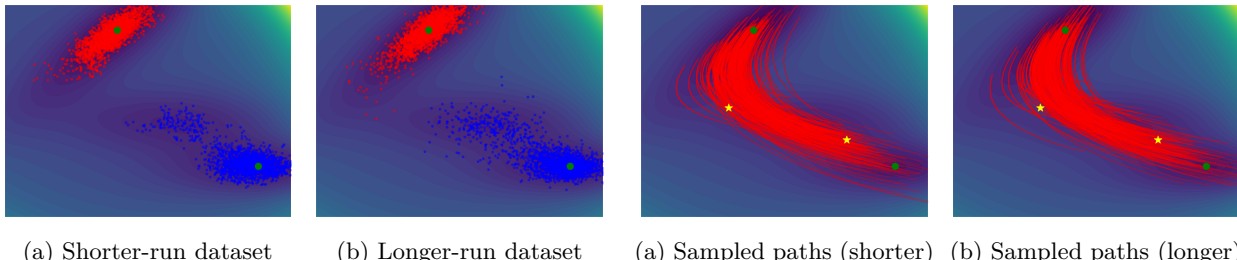

(a) Shorter-run dataset    (b) Longer-run dataset    (a) Sampled paths (shorter)   (b) Sampled paths (longer)

Figure 1: Both datasets in the figure contain 2000 pairs of data points, but randomly sampled from simulation of 4K and 12K steps, respectively.

Figure 2: Sampled paths from models trained on both the shorter-run and longer-run datasets (Saddle points are stared).

### 3.5 Resampling and Replay Buffer

Noting the fact that our coupling parameterizations above do not reflect the desired dynamical cost where $V(\cdot) := U(\cdot)$, we introduce a resampling procedure to reweight paths induced by our coupling $p_{0,T}$ and spline parameterization of $p_{t|0,T}^{\phi}$ or $x_t^{\phi}(x_0, x_T)$.

$$\tilde{w}(x|x_0, x_1) = \int_0^T \left( \frac{1}{2} \|v_\theta(x_t)\|^2 + U(x_t) \right) \mathrm{d}t \tag{20}$$

We aim to assign greater weights to the lower cost paths and smaller weights to the higher cost paths. We normalize the cost as weight $w_i = \exp(-\tilde{w}(x^i))/ \sum_{n=1}^{N} \exp(-\tilde{w}(x^i))$.

After sampling lower cost transition paths, we push them into a replay buffer $\mathcal{B}$ by their importance weights. In addition, we refine the learned neural spline by drawing samples from the replay buffer. Additional sampling scheme can be developed for replay buffer such as sampling without replacement Feng et al. (2023). To do so, we sample paths $\gamma = \{x_t\}_{t=0}^T$ with replacement and optimize the following objective to align the neural spline with the paths:

$$\mathcal{L}_{\text{replay}}(\phi) = \mathbb{E}_{\mathcal{U}(t)} \mathbb{E}_{p_{0,T}} \mathbb{E}_\gamma \|x_\phi(x_0, x_T, t) - \gamma_t\|^2 + \|v_\phi(x_0, x_T, t)\|^2 \tag{21}$$

In addition, we apply a kinetic energy loss function to ensure the smoothness of the learned spline.

## 4 Experiment

### 4.1 Experiment Set-up

**Müller-Brown Potential.** We first employ the Müller-Brown potential which is a commonly used mathematical model to study transition paths between metastable states. The energy landscape is characterized by three local minima and two saddle points connecting them and can be written down analytically in Appendix E.1. To simulate this system, we run the first-order Langenvin dynamics around each local minima.

$$x_{t+1} = x_t - \nabla_x V(x_t) \cdot dt + \sqrt{dt} \cdot \mathrm{diag}(\xi) \cdot \varepsilon, \quad \varepsilon \sim \mathcal{N}(0, 1) \tag{22}$$

where we apply Euler discretization of the continuous dynamics. In our experiment, we set $dt = 10^{-4}$ and $\xi = 5$.

**Alanine Dipeptide.** We validate our proposed method on a real-world molecular system, Alanine Dipeptide, which contains 22 atoms. The transition between two metastable states (C7eq and C7ax) is characterized by a two-dimensional free energy surface ($\phi$, $\psi$ dihedral angles). The molecular configurations are sampled by conventional molecular dynamics (cMD). MD simulations are performed in vacuum for 1.2 ns with 2 fs time step for each metastable state with the AMBER99SB-ILDN force field. Trajectories and CVs

| Method | Evaluations | MinMax Energy | Max Energy | Distance $d_1{}^2$ | Distance $d_2{}^3$ |
|---|---|---|---|---|---|
| MCMC | 1.03B | -40.21 | $-17.80 \pm 14.77$ | - | - |
| Doob's Lagrangian | 1.28M | -40.56 | $-14.81 \pm 13.73$ | - | - |
| Linear (Random) | N/A | -37.01 | $7.51 \pm 13.0$ | $0.62 \pm 0.16$ | $0.13 \pm 0.09$ |
| Linear (OT) | N/A | -38.98 | $5.22 \pm 17.82$ | $0.59 \pm 0.22$ | $0.20 \pm 0.14$ |
| Ours | 4K | -40.66 | $-27.66 \pm 20.99$ | $0.17 \pm 0.10$ | $0.15 \pm 0.13$ |
| Ours | 12K | -40.53 | $-27.30 \pm 9.85$ | $0.20 \pm 0.10$ | $0.17 \pm 0.17$ |
| Ours | 20K | -40.66 | $-31.93 \pm 9.12$ | $0.14 \pm 0.09$ | $0.15 \pm 0.11$ |

Table 1: Müller-Brown potential quantitative evaluation. 1,000 paths sampled from models trained on both datasets in Figure 1 and one additional longer (20K) simulation steps dataset. For each sampled 1,000 paths, we report the distribution of maximum energy state along the path and minimum energy of the maximum energy states, and the shortest distance points to the two saddle points. Visualization for linear interpolation can be found in Figure 5.

are recorded every 40 fs. Langevin integrator are used to maintain the system temperature to 300 K. HBonds were constrained during simulations.

The free energy surface (FES) is obtained by 800-ns well-tempered metadynamics (WT-MetaD) simulations. Two backbone dihedral angles are chosen as collective variables (CV). Each CV axis is evenly discretized with 25 grid points and the Gaussian bias potential are deposited every 2 ps along the CVs grids. The height and width of the bias potential are set to 0.2 kj/mol and 0.05 radians respectively. The 2D FES is collected and summed from the Gaussian deposits. FES is converged to 0.1 kcal/mol after 800 ns by examining the RMSE of FES between adjacent time stamps. To improve the convergence, we employ the well-tempered version of MetaD with a scaling factor of 8. The simulation is performed via the OpenMM[1] software.

To achieve translation and rotation equivariance, we use the internal coordinate system in addition to the Cartesian coordinate system, more details can be found in Appendix B.

**Hardware.** All experiments are conducted on two NVIDIA GeForce RTX 4090 GPU cards.

### 4.2   2D Toy Potential: Muller-Brown Potential

The Müller-Brown potential has three local minima and two saddle points following the closed-form potential energy surface in Equation (46). Starting from the initial and final local minima located at (-0.56, 1.44) and (-0.05, 0.47) in Figure 1, we generate the training data by simulating the first-order Langevin Dynamics Equation (22) for 4,000, 12,000, and 20,000 steps with $\mathrm{d}t = 10^{-4}$. As shown in Table 1, a minimum of 4,000 steps simulation of molecular dynamics around local minima can give us decent paths that are close to the saddle points. As the simulation increases to 12,000 and 20,000 steps, it further improves the performance. In addition, our method requires much fewer data compared to MCMC's 1.03B and Doob's Lagrangian's 1.28M simulation steps. It is worth noting that our methods cannot sample from the true transition path distribution as MCMC Dellago et al. (2002) and Doob's Lagrangian Du et al. (2024), but we show we can hit around the saddle points quite efficiently.

### 4.3   Molecular System: Alanine Dipeptide

In Figure 3, we visualize 50 randomly sampled transition paths from our method, we can observe that most of the paths find the correct collective variable $(\phi, \psi)$ dihedral angles in a much higher (66) dimensional space only by learning potential energy from a short-run molecular dynamics simulation around the local minima. In addition, the two ways of learning potential energy result in similar sampled path distributions. Nevertheless, we find the sampled transition paths with the flipped dihedral angles. In Figure 4, we demonstrate a qualitative showcase from a low-energy path, transitioning between the two metastable states.

---

[1]https://openmm.org/
[2]$d_1$ is the shortest distance to the first saddle point (-0.77, 0.64)
[3]$d_2$ is the shortest distance to the second saddle point (0.22, 0.3)

| Potential | Coord | Resampling | Coupling | # sample pairs | # evals | MinMax Energy | Max Energy | Run Time (min) |
|---|---|---|---|---|---|---|---|---|
| Linear | Cartesian | N/A | Product | N/A | N/A | 51,521.56 | 9.27E+21 ± 9.27E+22 | N/A |
| Linear | Cartesian | N/A | OT | N/A | N/A | 27,818.31 | 3.70E+17 ± 3.51E+18 | N/A |
| Linear | Internal | N/A | Product | N/A | N/A | 554.71 | 8.04E+16 ± 8.04E+17 | N/A |
| Linear | Internal | N/A | OT | N/A | N/A | 550.72 | 2.06E+12 ± 1.37E+13 | N/A |
| IDPP | Cartesian | N/A | Product | 100 | 200 | 893.98 | 26,325.18 ± 130,781.38 | N/A |
| IDPP | Cartesian | N/A | OT | 100 | 200 | 740.4 | 66,570.38 ± 254,968.37 | N/A |
| IDPP | Cartesian | N/A | Product | 2K | 4K | 2,186.33 | 140,672.19 ± 373,711.66 | N/A |
| IDPP | Cartesian | N/A | OT | 2K | 4K | 1,310.45 | 146,402.56 ± 401,000.31 | N/A |
| MCMC* | Cartesian | N/A | N/A | N/A | 38.40M | 821.20 | 821.29 ± 0.03 | N/A |
| MCMC | Cartesian | N/A | N/A | N/A | 1.29B* | 60.52 | 288.46 ± 128.31 | N/A |
| Metric | Cartesian | N/A | OT | 30K | 1.2M | 459.55 | 996.25 ± 389.94 | 18 |
| Metric | Internal | N/A | OT | 30K | 1.2M | 96.06 | 333.38 ± 267.013 | 20 |
| Latent | Cartesian | N/A | OT | 30K | 1.2M | 326.36 | 687.83 ± 226.78 | 17 |
| Latent | Internal | N/A | OT | 30K | 1.2M | 45.86 | 109.72 ± 69.24 | 25 |
| Latent | Cartesian | N/A | product | 30K | 1.2M | 1,098.04 | 2,632.22 ± 890.64 | N/A |
| Latent | Internal | N/A | product | 30K | 1.2M | 46.60 | 110.54 ± 73.25 | N/A |
| Metric | Cartesian | N/A | OT | 5K | 1.2M | 656.34 | 1505.19 ± 588.49 | N/A |
| Metric | Cartesian | N/A | OT | 10K | 1.2M | 554.71 | 1027.41 ± 338.83 | N/A |
| Metric | Cartesian | N/A | OT | 20K | 1.2M | 538.38 | 962.44 ± 340.89 | N/A |
| Metric | Cartesian | N/A | OT | 30K | 1.2M | 551.07 | 887.28 ± 259.34 | N/A |
| Metric | Cartesian | Resample-1 | OT | 30K | 16.2M | 345.39 | 593.28 ± 223.79 | N/A |
| Metric | Cartesian | Resample-2 | OT | 30K | 31.2M | 309.72 | 540.39 ± 193.49 | N/A |
| Metric | Cartesian | Resample-3 | OT | 30K | 46.2M | 382.28 | 661.68 ± 256.34 | N/A |
| Metric | Cartesian | N/A | Reflow-Product | 30K | 1.2M | 1,286.08 | 2,795.79 ± 1,150.02 | N/A |
| Metric | Cartesian | N/A | Reflow-1 | 30K | 1.2M | 693.74 | 1,336.80 ± 491.53 | N/A |
| Metric | Cartesian | N/A | Reflow-2 | 30K | 1.2M | 713.63 | 1,280.59 ± 388.75 | N/A |
| Metric | Cartesian | N/A | Reflow-3 | 30K | 1.2M | 694.59 | 1,367.04 ± 472.59 | N/A |
| Metric | Cartesian | N/A | Reflow-bi-Product | 30K | 1.2M | 897.21 | 2,897.45 ± 1,276.82 | N/A |
| Metric | Cartesian | N/A | Reflow-bi-1 | 30K | 1.2M | 874.77 | 2037.43 ± 798.60 | N/A |
| Metric | Cartesian | N/A | Reflow-bi-2 | 30K | 1.2M | 829.44 | 1,697.11 ± 637.72 | N/A |
| Metric | Cartesian | N/A | Reflow-bi-3 | 30K | 1.2M | 752.09 | 1,653.81 ± 551.88 | N/A |

Table 2: Alanine Dipeptide quantitative results. Top 100-weighted paths are evaluated for each setup with different ways of learned energies, coordinate systems, resampling steps and coupling parameterizations. The coupling parameterization includes product coupling, OT coupling, and both unidirectional and bidirectional (bi) reflow coupling. In addition, we report the training time for learning potential (latent) and kinetic energy (metric). (MCMC results are taken from Du et al. (2024), * indicates variable-length MCMC.)

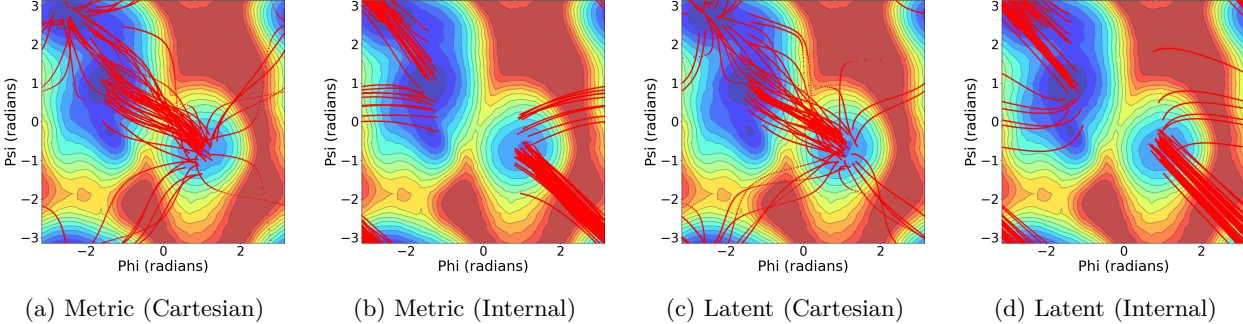

(a) Metric (Cartesian)  (b) Metric (Internal)  (c) Latent (Cartesian)  (d) Latent (Internal)

Figure 3: Alanine Dipeptide qualitative evaluation. 50 randomly sampled transition paths are shown for both parameterization in Cartesian and internal coordinate systems with two learned potential energies. Each models are trained over 30,000 data sampled uniformly from a 1.2ns simulation on each metastable states.

As shown in Table 2, the sampled paths by our method is way better than linear interpolating the two metastable states, in both Cartesian and internal coordinate systems. The commonly used image dependent pair potential method Smidstrup et al. (2014) (details in Appendix D) which interpolates in the pairwise distance matrices space perform better than linear interpolation but does not learn low-enery paths without distribution matching. In addition, we observe that the distributions of the maximum energy state over a path sampled by both approaches of learning potential energies are also close with the latent interpolation method being way faster. Next, we show how different sample sizes (5K, 10K, 20K and 30K) affect the

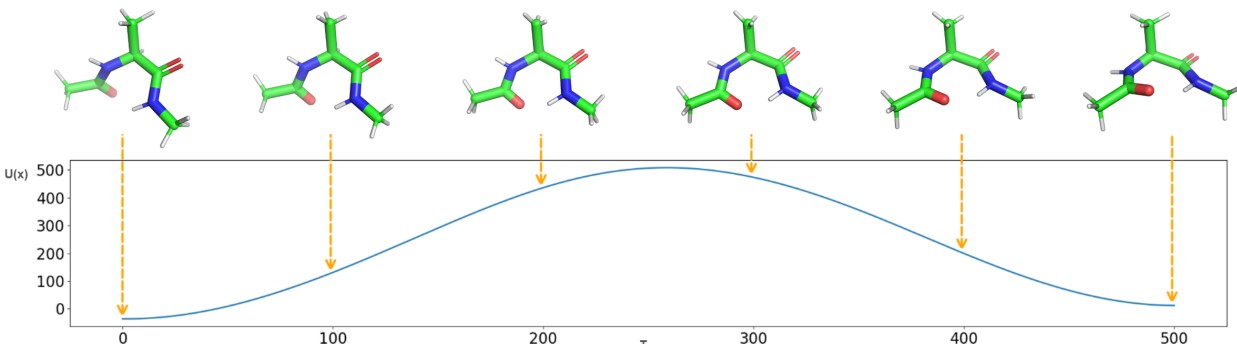

Figure 4: Alanine Dipeptide low-energy path visualization. A total of 500 timesteps from one metastable state to another going through an energy barrier.

quality of the sampled paths. In general, we find the performance improves with more data samples. We also compare different coupling parameterizations and we observe that the OT coupling is way better than the product measure as initial coupling. However, reflow effectively improves over the product coupling and approach the performance of the OT coupling. In the end, we validate the effectiveness of the resampling procedures. We find in general resampling improves the performance but saturates quickly after one iteration. It is also worth noting that resampling can be expensive as it requires calculating importance weights over an entire trajectory. Additional experimental results and training procedures can be found in Appendix F.

## 5 Conclusion, Limitation and Future Work

In this paper, we propose to simulate transition dynamics in molecular systems by inferring dynamics from the local dynamics around one metastable state to another. We propose a generalized flow matching algorithm that additionally optimizes a learned potential energy of the system. In our scenario, the potential energy is obtained through metric learning or latent space interpolation. We further employ an importance sampling technique and replay buffer to improve the convergence of the method. Experimental results demonstrate that the proposed method is capable of finding good transition paths, and good approximations of transition states with a significantly small number of energy evaluations.

One main limitation of the current framework is that the sampled path distribution could be far from the transition path distribution. One future direction is to use the learned path distribution as a proposal distribution for sampling-based approaches, e.g. Holdijk et al. (2024); another direction is to use the current framework as an initialization for transition state search methods.

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

# A    Additional Details on Metric Learning

Following Kapusniak et al. (2024), given a dataset of data samples $\{x_i\}_{i=1}^N$, we learn a metric from the Radial Basis Function (RBF) such that $G(x) = (\text{diag}(h(x)) + \epsilon I)^{-1}$, where $\epsilon > 0$ and the function $h(x)$ is defined as:

$$h(x) = \sum_{k=1}^K \omega_k \exp\left(-\frac{\lambda_k}{2}\|x - \bar{x}_k\|^2\right) \tag{23}$$

where $\bar{x}_k$ is the centroid of the $K$ clusters found by $k$-means clustering algorithm, $w_k$ is learned weights and $\lambda_k$ is a bandwidth associated with each cluster. The bandwidth around each cluster $C_k$ is defined as follows:

$$\lambda_k = \frac{1}{2}\left(\frac{\kappa}{|C_k|}\sum_{x \in C_k}\|x - \bar{x}_k\|^2\right)^{-2} \tag{24}$$

where $C_k$ is the $k$-th cluster, $\bar{x}_k$ is the centriod of the cluster, and $\kappa$ is a hyperparameter that controls the decay rate of the weight for cluster of different shapes. We then use the following objective function to learn the weights $w_k$ for each cluster in the dataset $\mathcal{D}$:

$$\mathcal{L}(\{\omega_k\}_{k=0}^K) = \sum_{x \in \mathcal{D}}(1 - h(x))^2 = \sum_{x \in \mathcal{D}}\left(1 - \sum_{k=1}^K \omega_k \exp\left(-\frac{\lambda_k}{2}\|x - \bar{x}_k\|^2\right)\right)^2 \tag{25}$$

# B    Translational and Rotational Invariance

A function $f$ is $G$-equivariant if $\forall x \in \mathbb{R}^d$, $g \in G$, we have $f \circ g(x) = g \circ f(x)$. A special case of equivariant function is the invariant function such that a function $f$ is $G$-invariant if $\forall x \in \mathbb{R}^d$, $g \in G$, we have $f \circ g(x) = f(x)$. The kinetic and potential energy of each state of a molecular system is invariant to the Euclidean group while the Cartesian coordinate and vector field are equivariant to the Euclidean group. We represent the molecular system in internal coordinates following Noé et al. (2019) instead of Cartesian coordinates which are constructed by invariant quantities, i.e. distances, angles and dihedral angles. In this scenario, we remove the unnecessary degrees of freedom, i.e. translations, rotations and reflections.

# C    Proofs

## C.1    Conditional Probability Paths

Toward deriving our conditional objective, we begin by showing the following lemma (Tong et al., 2023).

**Lemma 1** *For a given joint distribution $p_{0,T}$ and conditional distribution $p_{t|0,T}$ such that $p_t(x_t) = \mathbb{E}_{p_{0,T}}[p_{t|0,T}(x_t)]$, and a conditional vector field $v_{t|0,T}$ which satisfies the continuity equation,*

$$\frac{\partial}{\partial t}p_{t|0,T}(x_t) = -\nabla \cdot \left(p_{t|0,T}(x_t)\ v_{t|0,T}(x_t)\right), \tag{26}$$

*then, under mild conditions, the vector field*

$$v_t(x_t) = \mathbb{E}_{p_{0,T}(x_0,x_T)}\left[\frac{p_{t|0,T}(x_t)}{p_t(x_t)}v_{t|0,T}(x_t)\right] \tag{27}$$

*satisfies the continuity equation*

$$\partial_t p_t(x_t) = \mathbb{E}_{p_{0,T}}\left[\partial_t p_{t|0,T}(x_t)v_{t|0,T}(x_t)\right] = -\nabla \cdot \left(p_t(x_t)v_t(x_t)\right) \tag{28}$$

Assuming the Leibniz rule holds, we decompose

$$\frac{\partial}{\partial t}p_t(x_t) = \frac{\partial}{\partial t}\mathbb{E}_{p_{0,T}(x_0,x_T)}\left[p_{t|0,T}(x_t|x_0,x_T)\right] = \mathbb{E}_{p_{0,T}(x_0,x_T)}\left[\frac{\partial}{\partial t}p_{t|0,T}(x_t|x_0,x_T)\right] \tag{29}$$

We will introduce a vector field $v_{t|0,T}$ which is constrained to satisfy the continuity equation for $p_{t|0,T}$. Omitting explicit conditioning in the arguments, we write

$$\frac{\partial}{\partial t} p_{t|0,T}(x_t) = -\nabla \cdot \left( p_{t|0,T}(x_t) \, v_{t|0,T}(x_t) \right). \tag{30}$$

Finally, we would like to relate the conditional vector field $v_{t|0,T}$ to the marginal vector field $v_t$ in eq. (35). Following Tong et al. (2023) Thm 3.1, we confirm that

$$v_t(x_t) = \mathbb{E}_{p_{0,T|t}(x_0,x_T|x_t)} \left[ v_{t|0,T}(x_t) \right] = \frac{1}{p_t(x_t)} \mathbb{E}_{p_{0,T}(x_0,x_T)} \left[ p_{t|0,T}(x_t) v_{t|0,T}(x_t) \right] \tag{31}$$

satisfies the continuity equation relationships in eq. (29)-eq. (30), namely

$$\frac{\partial}{\partial t} p_t(x_t) = \mathbb{E}_{p_{0,T}(x_0,x_T)} \left[ \frac{\partial}{\partial t} p_{t|0,T}(x_t) \right] = \mathbb{E}_{p_{0,T}(x_0,x_T)} \left[ -\nabla \cdot \left( p_{t|0,T}(x_t) \, v_{t|0,T}(x_t) \right) \right] \tag{32}$$

$$= -\nabla \cdot \mathbb{E}_{p_{0,T}(x_0,x_T)} \left[ p_{t|0,T}(x_t) \, v_{t|0,T}(x_t) \right] \tag{33}$$

$$= -\nabla \cdot \left( p_t(x_t) v_t(x_t) \right) = \frac{\partial}{\partial t} p_t(x_t) \tag{34}$$

where, in the second line, we use the linearity of divergence operator and expectation to swap their order and, in the third line, we substitute the identity in eq. (31) to recover $\frac{\partial}{\partial t} p_t(x_t)$.

## C.2 Proof of Conditional Objective

We begin from the Fokker-Planck equation formulation of the GFM objective

$$\mathcal{L}_{\text{GFM}} = \min_{v_t, p_t} \int_0^T \int \left( \frac{1}{2} \|v_t(x_t)\|^2 + V_t(x_t) \right) p_t(x_t) \mathrm{d}x \mathrm{d}t \tag{35}$$
$$\text{s.t.} \quad \partial_t p_t(x_t) = -\nabla \cdot \left( p_t(x_t) v_t(x_t) \right), \; p_0 = \mu_0, \; p_T = \mu_T$$

Assume that a path of marginals $p_t(x_t)$ can be decomposed as $p_t(x_t) = \mathbb{E}_{p_{0,T}}[p_{t|0,T}(x_t)]$, where we assume the joint distribution $p_{0,T} \in \Pi(\mu_0, \mu_T)$ satisfies the endpoint constraints and the conditional distribution $p_{t|0,T}$ is suitably smooth (e.g. absolute continuity) such that there exists $v_{t|0,T}$ satisfying $\partial_t p_{t|0,T} = -\nabla \cdot \left( p_{t|0,T} v_{t|0,T} \right)$. Under these assumptions, we show that the objective in eq. (35), as a function of $p_t, v_t$, can be upper bounded in terms of the conditional $p_{t|0,T}, v_{t|0,T}$. [4]

For any $p_t, v_t$ satisfying the above, we begin by rewriting the continuity equation constraint

$$\partial_t p_t(x_t) = \mathbb{E}_{p_{0,T}(x_0,x_T)} \left[ \partial_t p_{t|0,T}(x_t) \right] = \mathbb{E}_{p_{0,T}(x_0,x_T)} \left[ -\nabla \cdot \left( p_{t|0,T} v_{t|0,T} \right) \right] \tag{36}$$

where assumed $p_{0,T}$ satisfies the endpoint marginal constraints.

Turning to the objective, we write the expectation using $\mathbb{E}_{p_t}[V_t(x_t)] = \mathbb{E}_{p_{0,T}} \mathbb{E}_{p_{t|0,T}}[V_t(x_t)]$,

$$\int_0^T \int \left( \frac{1}{2} \|v_t(x_t)\|^2 + V_t(x_t) \right) p_t(x_t) \mathrm{d}x \mathrm{d}t \tag{37}$$

$$= \int_0^T \int \frac{1}{2} \|v_t(x_t)\|^2 p_t(x_t) dx_t dt + \mathbb{E}_{p_{0,T}(x_0,x_T)} \left[ \int_0^T \int V_t(x_t) p_{t|0,T}(x_t) dx_t dt \right] \tag{38}$$

---

[4]We assume the decomposition into suitable $p_t(x_t) = \mathbb{E}_{p_{0,T}}[p_{t|0,T}(x_t)]$ does not change the value of the optimization. Otherwise, a further bound would be induced.

Finally, we would like to express the $v_t$ term in terms of $v_{t|0,T}$ to match the constraint in (36). Using the identity eq. (31) and then Jensen's inequality, we write

$$= \int_0^T \int \frac{1}{2} \left\| \mathbb{E}_{p_{0,T|t}} \left[ v_{t|0,T}(x_t) \right] \right\|^2 p_t(x_t) dx_t dt + \mathbb{E}_{p_{0,T}} \left[ \int_0^T \int V_t(x_t) p_{t|0,T}(x_t) dx_t dt \right] \quad (39)$$

$$\leq \int_0^T \int \frac{1}{2} \mathbb{E}_{p_{0,T|t}} \left\| v_{t|0,T}(x_t) \right\|^2 p_t(x_t) dx_t dt + \mathbb{E}_{p_{0,T}} \left[ \int_0^T \int V_t(x_t) p_{t|0,T}(x_t) dx_t dt \right] \quad (40)$$

$$= \mathbb{E}_{p_{0,T}} \left[ \int_0^T \int \left( \frac{1}{2} \left\| v_{t|0,T}(x_t) \right\|^2 + V_t(x_t) \right) p_{t|0,T}(x_t) \ dx_t dt \right] \quad (41)$$

where, in the last line, we use the fact that $p_{0,T|t} p_t = p_{0,t,T} = p_{0,T} p_{t|0,T}$ for any $x_0, x_t, x_T$.

We have thus shown that any feasible, factorizable $p_t, v_t$ corresponds to an upper bound on the objective in terms of a coupling $p_{0,T}$ and conditional $p_{t|0,T}, v_{t|0,T}$. To formalize our conclusions, we define the conditional objective for particular $x_0, x_T \sim p_{0,T}$

$$\mathcal{L}_{\mathrm{cGFM}}(x_0, x_T) := \min_{p_{t|0,T}, v_{t|0,T}} \int_0^T \int \left( \frac{1}{2} \|v_{t|0,T}^\phi(x_t)\|^2 + V_t(x_t) \right) p_{t|0,T}(x_t|x_0, x_T) dx_t dt \quad (42)$$

$$\text{s.t.} \quad \partial_t \ p_{t|0,T}(x_t) = -\nabla \cdot \left( p_{t|0,T}(x_t) v_{t|0,T}^\phi(x_t) \right), \quad p_{0|0,T} = \delta_{x_0}, \quad p_{T|0,T} = \delta_{x_T}$$

where we abbreviate the boundary condition $p_{0|0,T}(x) = \delta(x - x_0)$.

Finally, we reintroduce the outer expectation over $p_{0,T}$ and consider optimizing over $p_{0,T} \in \Pi(\mu_0, \mu_T)$ (such that $p_0 = \mu_0, p_T = \mu_T$). We can thus conclude that optimizing the conditional objective upper bounds $\mathcal{L}_{\mathrm{GFM}}$, due to our use of decomposable $p_t$ and Jensen's inequality in equation 40

$$\mathcal{L}_{\mathrm{GFM}} \leq \min_{p_{0,T}} \mathbb{E}_{p_{0,T}} \left[ \mathcal{L}_{\mathrm{cGFM}}(x_0, x_T) \right] \quad \text{s.t.} \quad p_0 = \mu_0, \ p_T = \mu_T \quad (43)$$

## D   Image Dependent Pair Potential (IDPP) Method

In addition to linear interpolation, there is another class of methods based on interpolating the pairwise distance matrices of the two states to find a good initialization for transition paths. Similarly, we parameterize a neural spline $x^\phi$ to match the pairwise distance matrices given by the linearly interpolated pairwise distance matrices $r_t$ with the following objective:

$$r_t = (1-t)d(x_0) + td(x_T), \quad (x_0, x_T) \sim p_{0,T}, \ t \in [0, T] \quad (44)$$

$$\mathcal{L}_{\mathrm{IDPP}} = \sum_{i,j=1}^N \left( d(x_t^\phi) - r_t \right)^2 \quad (45)$$

where $d : \mathbb{R}^{N \times 3} \to \mathbb{R}^{N \times N}$ takes a Cartesian coordinate and returns the pairwise distance. We set the learning rate as 0.01 and train for 500 iterations in the experiment.

## E   Additional Experimental Details

### E.1   Müller-Brown Potential

The Müller-Brown potential can be written down analytically:

$$\begin{aligned} V(x, y) = &- 200 \exp \left( -(x-1)^2 - 10y^2 \right) \\ &- 100 \exp \left( -x^2 - 10(y - 0.5)^2 \right) \\ &- 170 \exp \left( -6.5(x + 0.5)^2 + 11(x + 0.5)(y - 1.5) - 6.5(y - 1.5)^2 \right) \\ &+ 15 \exp \left( 0.7(x + 1)^2 + 0.6(x + 1)(y - 1) + 0.7(y - 1)^2 \right) \end{aligned} \quad (46)$$

| Potential | Coord | Resampling | Coupling | # sample pairs | # evals | MinMax Energy | Max Energy |
|-----------|-------|------------|----------|----------------|---------|---------------|------------|
| Metric | Cartesian | N/A | Reflow-bs1 | 30K | 1.2M | 14,235.32 | 29,160.60 ± 7,273.21 |
| Metric | Cartesian | N/A | Reflow-bs1-bi | 30K | 1.2M | 23,833.99 | 35,034.83 ± 6,129.76 |
| Metric | Cartesian | N/A | Reflow-bs200-bi | 30K | 1.2M | 678.98 | 1,514.05 ± 280.01 |
| Metric | Cartesian | N/A | Reflow-bs1450-bi | 30K | 1.2M | 1,238.69 | 2,727.82 ± 790.01 |
| Metric | Cartesian | N/A | Reflow-%2 | 30K | 1.2M | 753.43 | 1788.25 ± 694.72 |
| Metric | Cartesian | N/A | Reflow-%5 | 30K | 1.2M | 922.77 | 1843.17 ± 722.33 |
| Metric | Cartesian | N/A | Reflow-%10 | 30K | 1.2M | 897.21 | 3600.76 ± 1237.46 |
| Metric | Cartesian | N/A | Reflow-%20 | 30K | 1.2M | 1129.51 | 3348.60 ± 1540.00 |
| Metric | Cartesian | N/A | Reflow-%2-bi | 30K | 1.2M | 936.38 | 2096.84 ± 793.43 |
| Metric | Cartesian | N/A | Reflow-%5-bi | 30K | 1.2M | 813.61 | 1882.41 ± 661.33 |
| Metric | Cartesian | N/A | Reflow-%10-bi | 30K | 1.2M | 544.59 | 1340.51 ± 462.89 |
| Metric | Cartesian | N/A | Reflow-%20-bi | 30K | 1.2M | 927.86 | 1508.29 ± 316.14 |

Table 3: Additional quantitative results for Alanine Dipeptide. Top 100-weighted paths are evaluated for each setup with different reflow coupling parameterizations. (bi means bidirectional reflow, bs means reflow every certain number of batches, and % denotes number of product coupling training before one reflow training.)

The potential energy function is trained for 400 epochs, neural spline and velocity networks are updated simultaneously and trained for 50 epochs, respectively with a batch size of 256. We use the Adam optimizer for all training with $10^{-2}$, $10^{-5}$ and $10^{-3}$, respectively. For each neural network, we use a three-layer MLP, with 128 hidden units per layer, and the SELU activation function. The clustering bandwidth for metric learning in Equation (24) is set to $\kappa = 1.5$. The number of clusters $K$ is set to 100 in Equation (23).

### E.2 Alanine Dipeptide

The potential energy function is trained for 400 epochs, and neural spline network and velocity network are trained for 50 epochs, respectively with a batch size of 512. We use the Adam optimizer for all training with $10^{-2}$, $10^{-5}$ and $10^{-3}$, respectively. For each neural network, we use a three-layer MLP, with 128 hidden units per layer, and the SELU activation function. The clustering bandwidth for metric learning in Equation (24) is set to $\kappa = 1.5$. The number of clusters $K$ is set to 150 in Equation (23). And in metric formula, we set $\epsilon$ to 0.001. To obtain the latent space for interpolation, we train a 32-dimension latent space VAE with a three-layer MLP encoder with 64 hidden units and anothor three-layer MLP decoder with 64 hidden units. Note the learning rate is $10^{-3}$ instead of $10^{-2}$. For all experiments, we use the scale of 0.1 Angstrom as unit length.

## F   Additional Training Procedure

**Varying training iterations for reflow.** We further explore how much the performance degrades when we shift to an online reflow procedure. In Table 3, the first part demonstrates when we reflow every single iteration, the performance drops significantly, potentially attributing to the error accumulation in each step. However, we notice the performance increases when we reflow after a decent amount of iterations. In the second part, we employ an additional training scheme where we iterate between the product coupling and reflow training. Surprisingly, we find that even if we still mainly train over the product coupling, incorporating reflow every several step improves the performance.

**Separate training of spline and velocity networks.** In Algorithm 1, we update the spline network and the velocity network simultaneously. Here, we also adopt a different training procedure, where the spline network is trained first, followed by the training of the velocity network. Apart from training the spline and velocity networks separately for 100 epochs instead of 50 epochs, all model and experimental setups remain consistent with those used in Table 2. The results are shown in Table 4.

| Potential | Coord | Resampling | Coupling | # sample pairs | # evals | MinMax Energy | Max Energy | Run Time (min) |
|---|---|---|---|---|---|---|---|---|
| Metric | Cartesian | N/A | OT | 30K | 1.2M | 527.94 | 953.39 ± 376.24 | 100 |
| Metric | Internal | N/A | OT | 30K | 1.2M | 36.43 | 103.49 ± 51.26 | 112 |
| Latent | Cartesian | N/A | OT | 30K | 1.2M | 376.26 | 606.41 ± 183.73 | 14 |
| Latent | Internal | N/A | OT | 30K | 1.2M | 35.89 | 104.21 ± 57.51 | 20 |
| Latent | Cartesian | N/A | Product | 30K | 1.2M | 573.94 | 1,813.98 ± 644.07 | N/A |
| Latent | Internal | N/A | Product | 30K | 1.2M | 27.81 | 94.43 ± 41.82 | N/A |
| Metric | Cartesian | N/A | OT | 5K | 1.2M | 798.15 | 1,421.74 ± 458.47 | N/A |
| Metric | Cartesian | N/A | OT | 10K | 1.2M | 502.75 | 877.29 ± 242.09 | N/A |
| Metric | Cartesian | N/A | OT | 20K | 1.2M | 527.72 | 943.12 ± 328.44 | N/A |
| Metric | Cartesian | N/A | OT | 30K | 1.2M | 527.95 | 953.39 ± 376.24 | N/A |
| Metric | Cartesian | Resample-1 | OT | 30K | 16.2M | 358.45 | 637.81 ± 183.07 | N/A |
| Metric | Cartesian | Resample-2 | OT | 30K | 31.2M | 310.61 | 586.98 ± 162.66 | N/A |
| Metric | Cartesian | Resample-3 | OT | 30K | 46.2M | 313.32 | 594.63 ± 160.11 | N/A |
| Metric | Cartesian | N/A | Product | 30K | 1.2M | 1,288.12 | 2,635.59 ± 1,066.55 | N/A |
| Metric | Cartesian | N/A | Reflow-1 | 30K | 1.2M | 875.42 | 2,004.55 ± 797.18 | N/A |
| Metric | Cartesian | N/A | Reflow2 | 30K | 1.2M | 958.48 | 2,119.66 ± 783.24 | N/A |
| Metric | Cartesian | N/A | Reflow-3 | 30K | 1.2M | 741.08 | 1,683.77 ± 620.79 | N/A |
| Metric | Cartesian | N/A | Reflow-bi-1 | 30K | 1.2M | 724.90 | 1,318.97 ± 463.80 | N/A |
| Metric | Cartesian | N/A | Reflow-bi-2 | 30K | 1.2M | 582.34 | 1,336.21 ± 503.13 | N/A |
| Metric | Cartesian | N/A | Reflow-bi-3 | 30K | 1.2M | 659.96 | 1,351.85 ± 533.67 | N/A |

Table 4: Alanine Dipeptide quantitative results with separate updates to dual networks. Top 100-weighted paths are evaluated for each setup with different ways of learned potential energies, coordinate systems, resampling steps and coupling parameterizations. The coupling parameterization includes product coupling, OT coupling, and both unidirectional and bidirectional (bi) reflow coupling. In addition, we report the training time for learning potential (latent) and kinetic energy (metric).

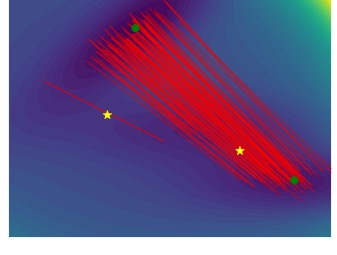 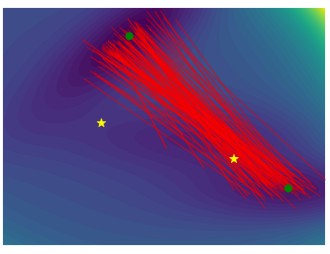

(a) Linear with OT  (b) Linear without OT

Figure 5: Linear interpolation paths for the Longer-run dataset with and without OT. Saddle points are labeled. Only 100 paths are selected.

