# OpenReview forum: "Generalized Flow Matching for Transition Dynamics Modeling"
_TMLR — Rejected by TMLR_

### Review · Reviewer_NSK4 · 2025-02-19

**Summary Of Contributions:**

- Novel Contribution to Flow Matching: The introduction of a potential energy term in the flow matching framework is a meaningful extension that improves transition path sampling.

- Computational Efficiency: Compared to MCMC and Doob's h-transform methods, this approach significantly reduces the number of expensive energy evaluations while still achieving high-quality transition paths.

- Theoretical Soundness: The framework is rigorously derived from Schrödinger bridge problems and optimal transport theory, with well-structured mathematical formulations.

- Practical Applicability: The approach is validated on both synthetic potentials (MüllerBrown) and real-world molecular simulations (Alanine Dipeptide), demonstrating its relevance to computational chemistry and molecular dynamics.

Resampling and Replay Buffer Strategy: The use of importance sampling and path replay buffers enhances learning efficiency and improves path quality over iterations.

**Audience:**

Yes

**Broader Impact Concerns:**

- Potential for Bias in Learned Energy Landscapes: The method learns a surrogate potential energy, which may introduce biases if the local simulations used for training are not sufficiently diverse. This could lead to incorrect transition pathways in high-stakes applications such as drug discovery or protein folding predictions.

- Generalization to Other Systems: The paper primarily evaluates small molecular systems. If applied to large-scale biological systems, the learned potential energy approximations may not generalize well, potentially leading to misleading conclusions.

- Ethical Considerations in Molecular Simulations: If used in drug design or chemical safety assessment, errors in the learned transition paths could lead to incorrect predictions about molecular stability or reaction pathways, impacting downstream applications.

Suggestion: The authors should include a Broader Impact Statement addressing these potential concerns, particularly the bias in learned energy landscapes and generalization limits to larger systems.

**Claims And Evidence:**

Yes

**Requested Changes:**

1. Clarify the Approximation Limitations
- Discuss the extent to which the sampled paths differ from the true transition path distribution.
- Compare the learned transition path probabilities with MCMC-based ground truth samples.

2. Improve Scalability Analysis
- Provide insights into how the approach scales to larger biomolecules or more complex free energy landscapes.

3. More Intuitive Explanation of Learned Potential Energy
- While the paper presents two methods (metric learning \& latent interpolation) for learning the potential energy, it would help to provide a clearer intuition or visualization of their differences and impact on transition paths.

4. Ablation Studies on Path Quality
- Conduct an ablation study by comparing different resampling strategies (e.g., varying replay buffer sizes) to better understand their contribution to performance improvements.

5. Address Sensitivity to Initialization
- Provide empirical results or insights on how different short-run molecular simulations affect the learned transition paths.

6. Add Missing Related Literatures
- DIFFMD: a geometric diffusion model for molecular dynamics simulations. AAAI 2022
- Geometric Trajectory Diffusion Models. NIPS 2024.
- Score Dynamics: Scaling Molecular Dynamics with Picoseconds Time Steps via Conditional Diffusion Model. JCTC 2024

**Strengths And Weaknesses:**

- Approximate Path Distribution: Unlike MCMC methods, the sampled paths may not fully capture the true transition path distribution, limiting its applicability in situations requiring exact probability distributions.

- A Sensitivity to Initialization: The performance of the learned potential energy function depends heavily on the choice of the initial short-run molecular simulations, which may introduce biases.

- Computational Trade-offs: While the method is more efficient than MCMC, it still requires a non-trivial amount of neural network training and parameter tuning for optimal performance.

- Limited Discussion on Scalability: The paper does not extensively discuss how well the method scales to higher-dimensional molecular systems beyond Alanine Dipeptide.

---

### Review · Reviewer_9n4z · 2025-03-05

**Summary Of Contributions:**

This paper tackles the problem of sampling transition paths between two metastable states of a dynamical system, which has applications particularly in molecular dynamics simulations. The paper proposes a generalised flow matching model and algorithm. The method extends flow matching by introducing an additional learnable surrogate energy function, drawing inspiration from the generalised Schrödinger bridge model. The algorithm relies on obtaining two boundary distributions via local simulations, which are used to infer the energy function through interpolation or metric learning. The paper evaluates the method on the Müller-Brown potential energy surface and the two-dimensional energy surface yielded by molecular dynamic simulations of the alanine dipeptide with two rotatable torsion angles.

**Audience:**

No

**Broader Impact Concerns:**

The paper does not include any statements of the broader impact. While I do not foresee any particular concern or ethical implication, besides the general concern of developing algorithms that can be applied for molecular simulations and drug developments, I do encourage the authors to reflect upon the potential broader impact and include a statement of their conclusions in the paper.

**Claims And Evidence:**

No

**Requested Changes:**

The most important changes would be, in my opinion:

- Improving the clarity of the exposition of the evaluation (section 4), as discussed in the previous section of my review.
- Consider incorporating additional, stronger baselines.
- Consider adding results in more challenging (higher-dimensional) problems.
- Fix citations
- Harmonise bibliography
- Discuss novelty

**Strengths And Weaknesses:**

### Strengths

The paper is generally well written and is easy to follow. In particular, the problem statement and motivation in the introduction are clearly explained and the paper provides sufficient background of the related work to facilitate the understanding of the method and algorithm presented in the paper. I found it helpful that the paper succinctly first describes diffusion models, flow matching and generalized Schrödinger bridge matching, while discussing some of their connections, applications and limitations. And that this background lays the basis for the specific method presented in the paper.

In terms of clarity, I also appreciate that the mathematical descriptions and derivations in Section 3 contains sufficient detail---as opposed to leaving the details for the supplementary material---and that the description of the algorithm in Algorithm 1 helps bind the multiple pieces together.

### Weaknesses

#### Evaluation

In my opinion, one of the most important weaknesses of the paper is the limited evaluation of the method and the poorer clarity of the evaluation section (Section 4) with respect to the rest of the paper. Regarding the breadth of the evaluation, the paper presents results on two rather simple or toy problems: the Müller-Brown potential energy surface and energy landscape of the alanine dipeptide. Both problems are two-dimensional and thus relatively simple, so it is unclear how well the method would scale to harder, higher-dimensional problems. While the alanine dipeptide problem is a molecule that indeed represents one of the main applications of transition dynamics modelling, it is generally used as a rather toy model, which can be tackled by relatively well-established methods.

Furthermore, the limited amount of baselines used for comparison also hinders the ability to assess the effectiveness of the method, in particular the claim that it offers a "low-cost approximation of the transition paths distribution" with "significantly small number of energy evaluations". On the Müller-Brown problem, the results in Table 1 show that the presented method uses considerably fewer evaluations than the two baselines---MCMC and Doob's Lagrangian---but I wonder to what extent the experiments are comparable, since the two baselines are meant to sample from the true transition path distribution, while the presented method samples from an approximation of the distribution. As a matter of fact, the metric Max Energy is much better for the baselines.

The results on the alanine dipeptide problem are much harder to analyses because they are not presented clearly enough (see below) but also because of the lack of baselines. For example, why is Doop's Lagrangian not included in this case? How to interpret the fact that IDPP uses many more evaluations than the presented method?

In terms of baselines, the paper cites several recent works that present models that could probably serve as baselines. For instance, the work by Holdijk et al. (2024), Yan et al. (2022), Seong et al. (2024), etc.

The clarity of the evaluation is hindered by several factors, in my opinion. First, Table 2 is very hard to interpret, as it contains a large number of numbers across many rows and columns that are difficult to digest. I would suggest to use visualisation techniques to plot the results and facilitate the interpretation. Also, why is the run time (last column) not provided for the vast majority of rows? Furthermore, the clarity of the writing in the section could be improved. For example, the metrics used for the evaluation are not introduced or defined in the text. The text claims that "our method is way better" than other methods but it is unclear under what criteria this statement holds. Neither are the baselines described in the text. By way of illustration, Table 2 uses the acronym IDPP for a method, which is not mentioned in the text (one can infer that it refers to the Image Dependent Pair Potential, mentioned in Appendix D). Finally, the discussion of the results and the influence of the various methods tried is rather vague and superficial, for instance towards the end of Section 4. To mention a couple of examples: only a couple of sentences are used to "analyse" the various coupling parameterisations used and no discussion is provided about why Cartesian and internal coordinates present so qualitatively different results in Figure 3. In my opinion, the paper could be largely improved by refining the structure and writing of Section 4.

#### Clarity and writing

There are several aspects regarding the clarity and the writing that can, in my opinion, be improved. One important issue that is present throughout the paper is that almost all if not all citations are in textual format---Einstein (1905)---even though many of them should be in parenthetical format---(Einstein, 1905). In other words, I recommend the authors to carefully revise when citations in LaTeX should be introduced with `\citet{}` (textual citation) and when with `\citep{}` (parenthetical citation). For further reference, see: https://www.overleaf.com/learn/latex/Natbib_citation_styles. Regarding citations, I would also consider avoiding the use of so-called _shotgun citations_, that is sets of many citations in one go, as is the case of the very first citation in the introduction, which references five papers. A better practice, in my opinion, would be to indicate which paper refers to which application, instead of leaving it to the reader to infer.

The bibliography does not follow any consistent format. For example, consider the two citations by Neklyudov et al. (2023, 2024). Both are at ICML in consecutive years, but the way their format is completely different.

There are a few acronyms and concepts that are used without prior definition, whose definition would improve the readability. For example, in page 4, I would mention that $\Pi$ indicates the joint distribution; in page 6, the concept of "dynamical cost" and the acronym "OT"; in page 7, "CVs".

Further, note that no punctuation is used after any equations. For example, there should be a comma after equation 2 and a period after equation 3, and so on.

Also, note that Algorithm 1 is never referenced in the main text.

Finally, here is a non-exhaustive list of possible typos or errors:

- Page 2: "which introduces an additional potential energy as constraint than the normal flow matching setup": grammar seems incorrect
- Page 2: "a vector field to moving from" -> "a vector field to move from"
- Page 2: "Wiener process" -> "Wiener processes"
- Page 3: "the the" -> "the"
- Page 3: "two metastable state" -> "two metastable states"
- Page 3: "to learn a surrogate potential energy functions" -> "to learn surrogate potential energy functions"
- Page 4: "that the latent space compresses semantic information from data thus better measure distance": "thus" does not seem like the right word.
- Page 4: "is a deterministic analogue the conditional stochastic control objective" -> "is a deterministic analogue of the conditional stochastic control objective"
- Page 5: "requires p and v which satisfy" -> "requires p and v to satisfy"
- Page 5: "we minimize the following the objective" -> "we minimize the following objective"
- Page 7: colon instead of period at the end of first paragraph of section 3.5
- Page 7: "sampling scheme" -> "sampling schemes"
- Page 7: "Alanine Dipeptide" -> "alanine dipeptide"
- Page 8: "The objective in section 3.4" -> "The objective in section 3.2"
- Page 8: "Langevin integrator are used" -> "A Langevin integrator is used"
- Page 8: "sampled transition paths from our method, we can observe" -> "sampled transition paths from our method. We can observe"
- Page 9: "the sampled paths by our method is way better" -> "the sampled paths by our method are way better" (plus, the tone sounds informal)
- Page 9: "low-enery" -> "low-energy"
- Page 16: "another" -> "another"

#### Other weaknesses

In my opinion, the paper would benefit for a clarification of its novel contributions. In particular, while it seems to be claimed (including in the title, and in section 3.2) that the paper introduces Generalized Flow Matching, in section 3.4 it is admitted that "[t]he objective in section 3.4 (sic) is a deterministic analogue the conditional stochastic control objective in Prop. 2 of Liu et al. (2024)". I think it would be important to better clarify what the novel contributions are specifically.

The additional results in Appendix F are included without providing any analysis or discussion.

---

### Review · Reviewer_8hgf · 2025-03-24

**Summary Of Contributions:**

In this paper, the authors formulate a generalized flow-matching framework that learns a vector field to sample probable paths connecting metastable states.  In other words, the goal is to find a low-cost approximation of the transition paths distribution, as simulating transition dynamics between metastable states is a fundamental challenge in dynamical systems in different domains ranging from physics to chemistry.

**Audience:**

Yes

**Broader Impact Concerns:**

N.A.

**Claims And Evidence:**

No

**Requested Changes:**

First, I believe a thorough round of proofreading is crucial. This should specifically address the following issues: correcting typos and improving the writing style, introducing any unspecified methods and notations, and clearly highlighting novelties as well as differences and similarities to previous works in a dedicated section.

Second, I believe the experiments section could be improved. At present, the results are difficult to navigate, particularly in the tables, which present an excessive amount of numbers and details, making it challenging to convey a clear message. Additionally, I do not observe any comparisons with other flow-matching or diffusion-based approaches. While I am not entirely familiar with the field, alanine dipeptide is a widely used benchmark. Therefore, I believe there are numerous methods in the literature that could serve as appropriate baselines.

Third, I found the duplicate sections describing the experimental setup and results (for both the Muller-Brown and alanine dipeptide systems) unnecessary and potentially confusing. Merging these into a single section per experiment type could streamline the presentation and improve the paper's overall accessibility.

As a minor point, I suggest fixing the citation style. Inline references should be enclosed in parentheses when they are not part of the main text.

I also wonder whether the work discussed in [this reference](https://openreview.net/forum?id=pRCOZllZdT&referrer=%5Bthe%20profile%20of%20Simon%20Olsson%5D(%2Fprofile%3Fid%3D~Simon_Olsson1)) may be related in scope to the paper under review. Specifically, the authors use Boltzmann Generators  [Noé et al.](https://www.science.org/doi/10.1126/science.aaw1147) to accelerate molecular dynamics (MD) between non-equilibrium and equilibrium states. Although this paper is not cited in the work under review, I find the scope to be somewhat similar, and I would appreciate it if the authors could comment on any differences and/or similarities between the two approaches.

**Strengths And Weaknesses:**

### Strengths
The problem addressed by the paper is highly relevant and of significant importance. Developing a low-cost approximator of the transition path probability distribution could have a substantial impact across various fields.

The appendix, although only inspected at a high level, appears to provide important theoretical results that complement the findings presented in the main text.

### Weaknesses
In its current form, the manuscript is not sufficiently polished for publication. Both the writing and the editing require additional rounds of refinement. I found the manuscript difficult to follow at times, with concepts and variables not being clearly defined. This lack of clarity makes it challenging for readers who are not thoroughly familiar with the flow-matching literature to engage with the paper.

The mathematical details often dominate the narrative, which may cause readers to lose track of the paper's core objectives and contributions. Although the mathematics appears to be rigorous, it is not always easy to connect the presented formulae with the paper's goals and its novel aspects.

Regarding the paper's novelty, I find it to be rather limited. Including a bullet-point summary of the key contributions at the end of the introduction, along with a dedicated Related Work section, may help to make the paper's novel aspects more explicit and easier to identify.

---

### Review · Reviewer_uwVh · 2025-03-24

**Summary Of Contributions:**

I encourage the authors work in this area but find the current submission far from ready for publication in TMLR.

**Audience:**

No

**Broader Impact Concerns:**

No Impact concerns.

**Claims And Evidence:**

Yes

**Requested Changes:**

The paper could be made to address the weaknesses noted above.

**Strengths And Weaknesses:**

Strengths:
The paper addresses a difficult problem with important applications.  Some of the paper is well written.

Weaknesses:
(1) Limited methodological contribution.
The significance/novelty of the methodological contribution is not made clear as compared to, for example, to Du et al 2024.

(2) limited methodological contribution.
The abstract announces validation of the effectiveness of the method on a "real-world" application.  This is misleading.  The application is to Alanine dipeptide absent of solvent, which is a standard toy system used for evaluating molecular simulation methods.  Notably this problem could be solved using standard MD simulations and looking at segments of paths that transitions between modes (an even sufficiently common in previous published simulations that it could be compared to as a "ground truth" for this toy system.  Notably, no such comparison is made (or is explained if presented in the table).

The discussion of empirical results is also unsatisfying.  Claims like "our method is way better" (sic) are not explained with reference any specific interpretations of the results.

(3) Rough editing.
There are many typos that the authors will find if they read the paper sentence by sentence to copy edit.

Nit. It is standard to introduce display equations as a part of sentence as if spoken in natural language with normal punctuation.  E.g. punctuation should be used when an equation ends a sentence or section, rather than without punctuation as at the end of section 2.1.

---

### Comment · Reviewer_uwVh · 2025-03-11
**Preliminary work in an interesting direction that is not ready to be published due to (1) limited methodological contribution, (2) limited empirical exploration and (3) rough editing.**

I encourage the authors work in this area but find the current submission far from ready for publication in TMLR.

(1) Limited methodological contribution.
The significance/novelty of the methodological contribution is not made clear as compared to, for example, to Du et al 2024.

(2) limited methodological contribution.
The abstract announces validation of the effectiveness of the method on a "real-world" application.  This is misleading.  The application is to Alanine dipeptide absent of solvent, which is a standard toy system used for evaluating molecular simulation methods.  Notably this problem could be solved using standard MD simulations and looking at segments of paths that transitions between modes (an even sufficiently common in previous published simulations that it could be compared to as a "ground truth" for this toy system.  Notably, no such comparison is made (or is explained if presented in the table).

The discussion of empirical results is also unsatisfying.  Claims like "our method is way better" (sic) are not explained with reference any specific interpretations of the results.

(3) Rough editing.
There are many typos that the authors will find if they read the paper sentence by sentence to copy edit.

Nit. It is standard to introduce display equations as a part of sentence as if spoken in natural language with normal punctuation.  E.g. punctuation should be used when an equation ends a sentence or section, rather than without punctuation as at the end of section 2.1.

---

### Decision · Action_Editor_1P3k · 2025-04-23

**Recommendation:** Reject

**Comment:**

Unfortunately the work is not ready for publication. On one hand, it requires considerable editing. The presentation is lacking clarity, making the paper hard to read for those that are not very familiar with the topic studied. Additionally, the evaluation of the method is very limited, making it hard to draw conclusions about the work.

These two aspects alone make the paper unsuitable for publication. Reviewers seem to agree on these points in their reviews and there seems to be a straightforward consensus that the work is not ready.

**Audience:**

The topic is reasonable for TMLR and of interest to the community.

**Claims And Evidence:**

The work provides limited evidence, as highlighted by reviewer 9n4z in terms of evaluating the model on Muller-Brown potential energy and the energy landscape of alanine dipeptide, both toy problems on which is hard to draw insight whether the proposed methodology works.
The baseline considered are limited as well. This view is agreed with among reviewers.